# SARS-CoV-2 T-Cell Responses in Allogeneic Hematopoietic Stem Cell Recipients following Two Doses of BNT162b2 mRNA Vaccine

**DOI:** 10.3390/vaccines10030448

**Published:** 2022-03-14

**Authors:** Béatrice Clémenceau, Thierry Guillaume, Marianne Coste-Burel, Pierre Peterlin, Alice Garnier, Amandine Le Bourgeois, Maxime Jullien, Jocelyn Ollier, Audrey Grain, Marie C. Béné, Henri Vié, Patrice Chevallier

**Affiliations:** 1CHU Nantes, CRCINA, INSERM, CNRS, Université de Nantes, F-44000 Nantes, France; thierry.guillaume@chu-nantes.fr (T.G.); jocelyn.ollier@inserm.fr (J.O.); audrey.grain@chu-nantes.fr (A.G.); mariecbene@gmail.com (M.C.B.); henri.vie@inserm.fr (H.V.); 2Hematology Department, Nantes University Hospital, Nantes University, F-44000 Nantes, France; pierre.peterlin@chu-nantes.fr (P.P.); alice.garnier@chu-nantes.fr (A.G.); amandine.lebourgeois@chu-nantes.fr (A.L.B.); maxime.jullien@chu-nantes.fr (M.J.); 3Virology Department, Nantes University Hospital, Nantes University, F-44000 Nantes, France; marianne.coste@chu-nantes.fr; 4Hematology Biology Department, Nantes University Hospital, Nantes University, F-44000 Nantes, France

**Keywords:** COVID 19, vaccine, BNT162b2, SARS-CoV-2 mRNA, allogeneic hematopoietic stem cell transplantation, cellular immunity, humoral immunity, TNFα, IFNγ, CD4^+^ T cells, CD8^+^ T cells

## Abstract

Background: At variance to humoral responses, cellular immunity after anti-SARS-CoV-2 vaccines has been poorly explored in recipients of allogeneic hematopoietic stem-cell transplantation (Allo-HSCT), especially within the first post-transplant years where immunosuppression is more profound and harmful. Methods: SARS-CoV-2 Spike protein-specific T-cell responses were explored after two doses of BNT162b2 mRNA vaccine in 45 Allo-HSCT recipients with a median time from transplant of less than 2 years by using INF-γ ELISPOT assay and flow-cytometry enumeration of CD4^+^ and CD8^+^ T lymphocytes with intracellular cytokine production of IFN-γ and TNF-α. Results: A strong TNF-α^+^ response from SARS-CoV-2-specific CD4^+^ T-cells was detected in a majority of humoral responders (89%) as well as in a consistent population of non-humoral responders (40%). Conclusions: T-cells are likely to participate in protection against COVID-19 viral infection, even in the absence of detectable antibody response, especially in the first years post-transplant in Allo-HSCT recipients.

## 1. Introduction

Since its appearance in the late months of 2019, COVID 19 infection has been responsible for more than 5 million deaths worldwide. Thanks to the rapid development of anti-SARS-CoV-2 vaccines, it has however been possible to consider controlling this pandemic, especially in terms of avoiding severe forms of infection leading to intensive care [1,2] Immunocompromised patients were initially excluded from trials evaluating the safety and efficacy of vaccination. These patients are now prioritized as they are considered at higher risk of COVID 19 infection and disease progression, although it is not clear whether their mortality rate differs from that of the general population [3]. In addition, reduced vaccine immunogenicity is increasingly reported in this setting [4,5,6,7,8], except in recipients of allogeneic hematopoietic stem cells transplantation (Allo-HSCT), where specific anti-SARS-CoV-2 antibodies can be detected in around 80% of cases [9,10,11,12]. This is rather intriguing since weak antibody responses to other vaccines have been well-documented in Allo-HSCT recipients [13], notably during the influenza pandemic of 2009 [14]. This discrepancy can be possibly related to the design of these new vaccines, based on messenger RNA (mRNA) that deliver the genetic information to produce the antigen rather than the antigen itself [15].

Humoral responses are however depending on and complementing virus-specific cellular adaptive responses. The latter provide the help necessary for B-cells proliferation and differentiation through specific CD4^+^ T-helper cells. Cellular immunity is also dependent on CD8^+^ cytotoxic T-lymphocytes (CTL) eliminating infected cells and thereby limiting the production and dissemination of viral particles. SARS-CoV-2-specific CD4^+^ and CD8^+^ T-cells can be detected in 100% and 70% of COVID-19 convalescent patients, respectively [16]. Moreover, SARS-CoV-2-reactive CD4^+^ T-cells can also be detected in ∼40–60% of unexposed individuals suggesting cross-reactive T-cell recognition between SARS-CoV-2 and other types of coronaviruses [16]. The same 100% rate of specific CD4^+^ T-cell responses (with T-helper type 1 polarization) is observed in healthy individuals after vaccination with either of the two types of anti-SARS-CoV-2 mRNA vaccines (BNT162b1, Pfizer BioNTech and mRNA-1273, Moderna). However, specific CD8^+^ T-cell responses have only been observed with BNT162b1 for unknown reasons [15]. In a series of solid organ transplant recipients vaccinated with BNT162b1, 48% have been reported to develop anti-SARS-CoV-2 CD4^+^ T-cell responses [17].

In Allo-HSCT recipients, where the overall survival rate is estimated at 68% at 30 days for those who contract COVID-19 infection [18], cellular immunity acquisition has been documented mainly in patients far from the transplant (median > 30 months) with heterogeneous results comprised between 19% and 82.3% of the population after 2 vaccines [19,20,21].

Here, we explored SARS-CoV-2 Spike protein-specific T-cell responses after two doses of BNT162b2 mRNA vaccine in allo-HSCT patients with a median from transplant of less than 2 years by using INF-γ ELISPOT assays and flow-cytometry enumeration of CD4^+^ and CD8^+^ T lymphocytes intracellular cytokine production of IFN-γ and TNF-α.

## 2. Methods

### 2.1. Patients

In a recent observational monocentric study of 117 hematopoietic Allo-HSCT adult recipients, we have reported that 54% and 83% patients, respectively achieved a humoral response after one and two doses of BNT162b2 [12]. Here, we considered 45 patients from the same cohort with respectively acute myeloblastic leukemia (AML, *n* = 26) or myelodysplastic syndrome (MDS, *n* = 19) and 16 healthy controls. All participants provided informed consent and the study was approved by the Ethical Review Board of Nantes University Hospital (2021-H03).

### 2.2. Peripheral Blood Mononuclear Cell (PBMC) Isolation

Peripheral blood was collected on Ethylenediaminetetraacetic acid (EDTA). All participants provided informed consent and the study was approved by the Ethical Review Board of Nantes University Hospital. PBMC were isolated using Ficoll density gradient centrifugation (Eurobio, Les Ulis, France) and were frozen with Fetal Bovine Serum-10% and dimethylsulfoxide (DMSO). PBMC were thawed and rested overnight in complete culture medium (RPMI, FBS 10%, GlutaMAX (2 nM), Penicillin-Streptomycin (PS, 100 UI/mL)) (Gibco, Thermofisher, Saint Herblain, France). Immunophenotype was determined by flow cytometry with fluorochrome-conjugated monoclonal antibodies: Fixable Viability Stain-780, CD45-V500; CD3-BUV395, CD14-PE, CD19-BB515 (BD Biosciences, Fremont, CA, USA) and HLA-DR-APC (Biolegend, San Diego, CA, USA).

### 2.3. Peptide Pools

The peptide pools consisted of 15mers with an 11 amino acid overlap spanning the whole protein sequence of the SARS-CoV-2 Spike glycoprotein (Prot _S1; _S+ and _S PepTivator peptide pools), 43 peptides from EBV proteins (pepTivator EBV-consensus) and pepTivator CMV pp65 (all from Miltenyi Biotec, Bergisch Gladbach, Germany). All were used at the concentration of 1 μg/mL for ex-vivo stimulation of PBMCs for INF-γ ELISPOT assay and flow cytometry.

### 2.4. INF-γ ELISPOT Assay

PBMC (2 × 10^5^) in 100 µL medium were added to each well of Human ELISpot^PRO^ Kit plates (Mabtech 3420-2AST-10, Nacka Strand, Sweden). Cells were incubated with culture medium (RPMI, FBS 10%, GlutaMAX (2 nM), supplemented by penicillin-streptomycin (PS, 100 UI/mL, Gibco, negative control). The 3 peptide pools covered the SARS-CoV-2 Spike glycoprotein and EBV or CMVpp65. Peptides were added as 100 µL to each well and, when possible, in duplicate wells. Plates were incubated at 37 °C and 5% CO_2_ for 24 h, developed according to the manufacturer’s instructions and dried before spot-counting on a Bioreader 5000-pro-S (BIOSYS GmbH, Karben, Germany). The median background for the negative control was 0 SFU/2 × 10^5^ cells (range 0–5). Frequencies of spot forming units (SFU) were reported per 100 CD3^+^ T-cells, evaluated beforehand in each PBMC suspension.

### 2.5. Intracellular Cytokine Staining by Flow Cytometry

SARS-CoV-2 T cell Analysis Kits (PBMC) from Miltenyi Biotec were used to analyze CD4^+^ and CD8^+^ SARS-CoV-2-reactive T cells. Aliquotes of 1 × 10^6^ PBMCs per well, in flat-bottom 96-well plates, were incubated with the different peptide pools, 2 µL of sterile water as a negative control or Cytostim^®^ (Miltenyi Biotec) as a positive control of T-cell stimulation. PBMC were stimulated at 37 °C and 5% CO_2_ for 2 h, then Brefeldin A (2 µg/mL) was added in each well and the plates were incubated for 4 additional hours. After incubation, PBMCs were collected, washed and stained according to the manufacturer’s instructions of the SRAS-CoV-2 Prot_S T cell Analysis Kit. A minimum of 100,000 live CD3^+^ T-cells were acquired on a BD LSRFortessa™ Instrument (BD Biosciences) and results were analyzed using FlowJo v.10.7.1 software (FlowJo, BD LifeSciences, Franklin Lakes, NJ). Doublets, debris, and dead cells as well as CD14^+^ and CD20^+^ cells were excluded. After gating on CD3^+^ as well as CD4^+^ or CD8^+^ cells, respectively, the respective 3 cytokine expression profiles IFN-γ^+^/TNF-α^−^, IFN-γ^+^/TNF-α^+^ and IFN-γ^−^/TNF-α^+^ were assessed for specific anti-Spike T-cell subsets. Percentages obtained for the unstimulated condition were subtracted from those of stimulated conditions.

## 3. Results

### 3.1. Patients

Patients and controls characteristics are given in Table 1. The median time between Allo-HSCT and first vaccination was 19.5 months (range: 3–126). Fifteen patients (33%) were less than a year after transplantation at the time of vaccination. The median time between the first vaccine injection and T-cell response evaluation was 56 days. The majority of patients were free of immunosuppressive or chemotherapy treatment (*n* = 37, 80%). None of the patients had acute graft-versus-host disease (GVHD) at the time of analysis.

### 3.2. Humoral and Cellular Responses

After two doses, all vaccinated healthy donors became seropositive and developed a T-cell response to Spike peptide pools (median: 0.02 SFU/100 T-cells, range 0.008–0.065) (Figure 1A). Anti-Spike IgG were detectable in 78% of Allo-HSCT patients (*n* = 35/45) and an anti-Spike T-cell response with IFN-γ production was also observed in 78% of the patients. Among humoral responders (HR), 89% (*n* = 31/35) had a positive anti-Spike CD3^+^ T-cell response with a median of 0.02 SFU/100 T-cells (range 0.003–0.272). This frequency is similar to that measured for healthy donors (Figure 1A). Of note, for 8 patients, the T-cell response was higher than that of healthy controls (>0.08 SFU/100 T-cells) which is equivalent to more than 1 specific T-cell per microliter of peripheral blood.

Among the 10 non humoral responders (NHR), 4 (40%) had developed cellular immunity, including one with a very high CD3^+^ T-cell response (0.13 SFU/100 T-cells).

As control, analysis of the frequency of EBV specific T-cells in both healthy donors and patients revealed that Allo-HSCT recipients often presented with higher frequencies of EBV specific T-cells, suggesting an ongoing post-transplant EBV reactivation (Figure 1A). PBMC immunophenotypic analysis showed that CD3^+^ levels were lower in patients compared to controls but similar between HR and NHR. The latter moreover had very low frequencies of B-cells and, interestingly, a higher frequency of CD14^+^ monocytes with low/neg HLA-DR expression, potentially corresponding to myeloid-derived suppressor cells (MDSCs) (Figure 1B).

For the 17 patients with the highest frequencies of anti-Spike CD3^+^ T-cells (≥0.096 SFU/100 T-cells) and in 12 healthy donors, enumeration of Spike-specific CD3^+^, CD4^+^ and CD8^+^ T-cells by flow cytometry confirmed the presence of specific SARS-CoV-2 T-cells (Figure 2) with frequencies that correlated with those obtained with the INF-γ ELISPOT assay (data not shown). These analyses revealed that, in contrast to healthy patients, the group of Allo-HSCT patients developed a clear predominance of anti-Spike CD4^+^ T-cell responses (Figure 2B). This CD4 predominance was not due to an intrinsic CD8^+^ T-cell dysfunction since strong CD8^+^ T-cell responses were observed against EBV-specific antigens, as well as against the super-antigen (CytoStim) used as a positive control (Figure 2B). Moreover, the anti-Spike CD4^+^ T-cell response was characterized by a high proportion of cells with an IFN-γ^−^/TNF-α^+^ cytokine pattern (Figure 2C).

As expected, the 6 patients who developed neither humoral nor cellular response were within one year of transplantation (Figure 3A). In this cohort, 9 patients were under treatment including 5 for active chronic GVHD (cyclosporine *n* = 2, cyclosporine + corticosteroids *n* = 3), while one patient was under corticosteroids for a chronic rheumatic disease and another one received 5′ azacytidine for relapse prevention. Two early patients were on their way to stop cyclosporine. Similar to reported data, undergoing immunosuppressive therapy did not predict a poor cellular response as 6 of 9 on-treatment patients (67%) still were able to develop Spike-specific T-cell responses (*p* = 1) (Figure 3B) [19]. Conversely to what has been reported [22], no reactivation of chronic GVHD occurred in our series after the two vaccines and none of the patients had developed COVID-19 infection at the time of analysis.

The only factor predicting no cellular response was a lower delay between the graft and the first vaccine (median 320 days vs. 886 days, *p* = 0.008).

## 4. Discussion

Overall, these data show that after two doses of BNT162b2 mRNA vaccine, 78% of Allo-HSCT recipients developed an anti-Spike cellular immunity with a predominance of specific CD4^+^ T cells with an IFN-γ^−^/TNF-α^+^ cytokine production pattern. The vaccine cellular response rate in this cohort is in agreement with recently published data by Harrington et al. (82.3%, *n* = 17; months from transplantation, median 55, range 19–172) [21] but higher than those first published by Ram et al. (19%, *n* = 37, months from transplantation, median 32, range 3–263) [19] and Lindemann et al. (29%, *n* = 117, months from transplantation, median 31, range 5–391) [20], even though the methods used were similar (ELISpot and Spike-peptides).

Interestingly, anti-SARS-CoV-2-specific CD3^+^ T-cells could be detected in 40% of the subset of NHR patients. In kidney transplant recipients, protective CMV-specific T-cell immunity is frequently observed in the absence of detectable anti-CMV antibodies [23]. Here, whether and to what extent the T-cell response observed in patients can protect from SARS-CoV-2 infection remains to be determined. The well-known role of TNF-α in the control of viral infection [24] and the strong IFN-γ^−^/TNF-α^+^ response observed for SRAS-CoV-2-specific CD4^+^ T-cells described here would be in favor of such a possibility.

Compared to the 3 previous studies with T cell responses analyses mentioned above, our cohort reports on the largest series of patients vaccinated within the first year after transplant (*n* = 15), where anti-Spike antibody response rates were particularly low (47%). Interestingly 66% of them yet developed a T-cell response. Thus, in the absence of antibodies, these patients could nevertheless benefit from a protection against COVID-19 infection through their cellular immunity, liable to rapidly destroy infected cells before they sprout newly formed virions. These results are particularly interesting since recent studies have reported that SARS-CoV-2 anti-Spike T-cell responses, induced by vaccination, remain robust against the omicron variant [25,26]. Indeed, detailed analysis of cellular immune responses in these patients vaccinated early after Allo-HSCT is of high importance. The preservation of T-cell responses should be now evaluated after the third vaccine injection.

A clear predominance of anti-Spike CD4^+^ T-cell responses characterized by a high proportion of cells with an IFN-γ^−^/TNF-α^+^ cytokine pattern was observed in our cohort. This confirms results observed in both studies by Ram et al. [19] and Harrigtonet al. [21]. In the latter [21], a polyfunctional T cell response, with dual expression of more than one proinflammatory cytokine within the same cell, was also observed in 70.6% of the patients while more than 90% of reactive T cells expressing pro-inflammatory cytokines and co-expression of CD45RO, a surface protein marker for memory T cells. This is important as the waning of humoral immune responses has been well documented a few months after post-infection or after vaccination [27].

Factors associated with the absence of humoral responses have been also well described in the setting of allotransplant including low B lymphocytes count, time-interval from allotransplant <12 months, ongoing immunosuppressive treatments and male gender [19,20,28,29]. Regarding factors predicting cellular response, Lindemann et al. [20] demonstrated an impact of age and of the time point after transplantation and vaccination. In the Ram study [19], T cell responses were correlated with the CD4^+^/CD8^+^ ratio. We confirm here the absence of T-cell response in patients who are closer from the transplant.

## 5. Conclusions

In conclusion, after two doses of BNT162b2 vaccine, a strong IFN-γ^−^/TNF-α^+^ response by SARS-CoV-2-specific CD4^+^ T-cells can be detected in a majority of humoral HR (89%) and many NHR (40%) Allo-HSCT recipients. The latter may participate in protection against COVID-19 viral infection, even in the absence of detectable antibody response, especially in the first months post-transplant.

## Figures and Tables

**Figure 1 vaccines-10-00448-f001:**
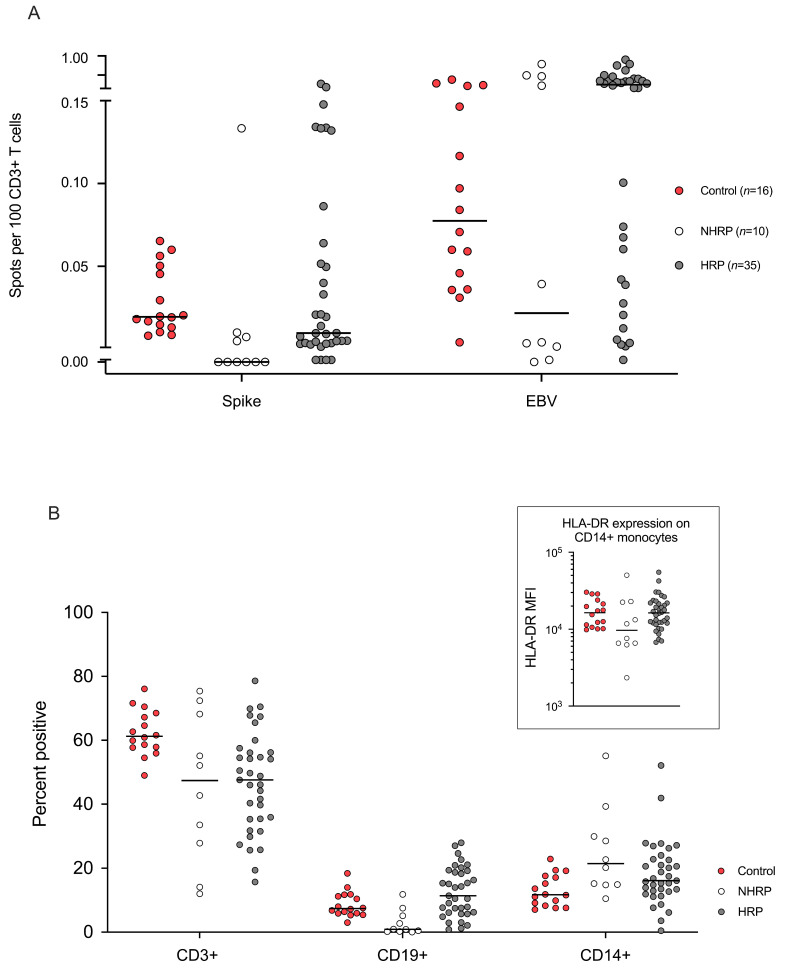
Anti-SARS-CoV-2 Spike T-cells analysis in 45 Allo-HSCT recipients (AML *n* = 26, MDS *n* = 19) with humoral (HRP *n* = 35) or no humoral (NHRP *n* = 10) response and healthy controls (*n* = 16) after two injections of the BNT162b2 mRNA vaccine. Panel (**A**) shows the number of IFNγ spots per 100 CD3^+^ T cells after stimulation of PBMC stimulated with Spike or EBV-consensus peptide; (**B**) PBMC phenotype analysis used for the ELISpot assay. Results show population frequencies among viable CD45^+^ cells. Horizontal lines indicate median values.

**Figure 2 vaccines-10-00448-f002:**
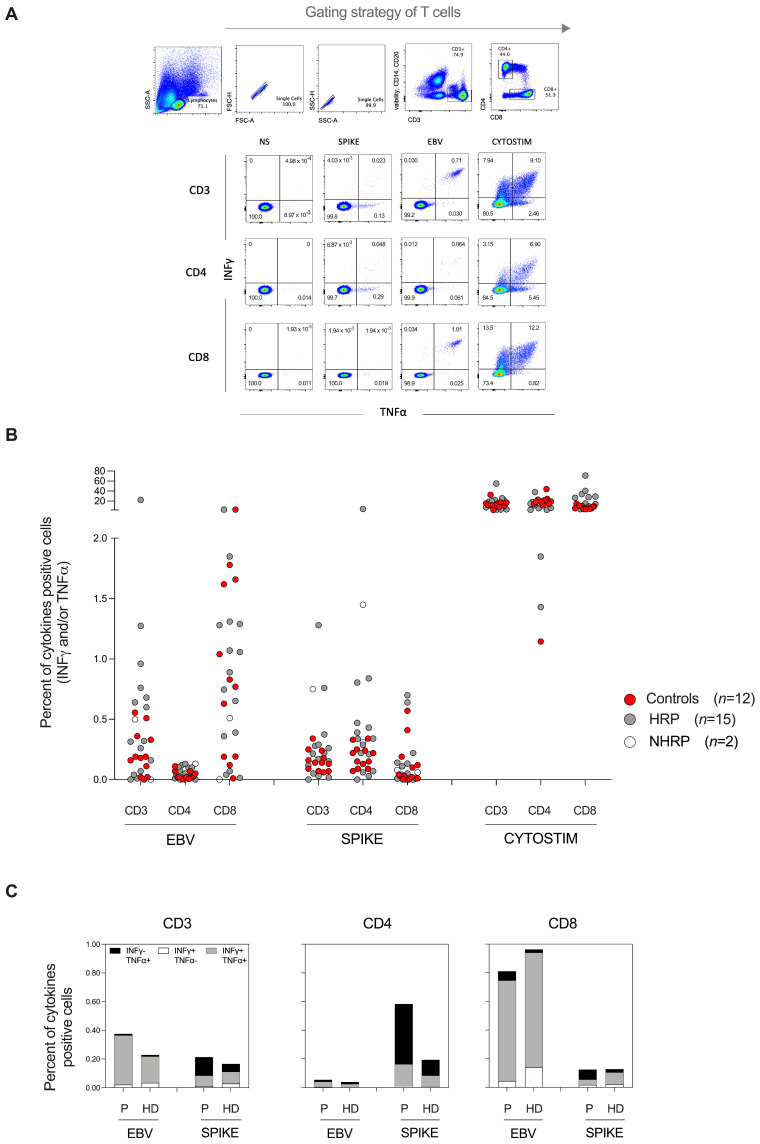
Features of specific CD3^+^, CD4^+^ and CD8^+^ T-cell responses against SARS-CoV-2 Spike and EBV peptides according to INFγ and TNFα production in Allo-HSCT recipients (*n* = 17: 2 NHRP, 15 HRP and 12 healthy donors (HD)) after two injections of BNT162b2 mRNA vaccine. PBMC were stimulated with Spike, EBV peptides, Cytostim (positive control) or not stimulated (NS). (**A**) Gating strategy and representative flow cytometry plots for one patient. (**B**) Dot plots representing the frequencies of CD3^+^, CD4^+^, and CD8^+^ T-cells producing IFN-γ, TNF-α, or both. Each dot represents one subject. For this group of patients, the magnitude of the INFγ^+^ CD3^+^ T-cell response correlated with that obtained by the INF-γ ELISPOT assay (data not shown). (**C**) Bar graphs showing the expression of INFγ and TNFα among SARS-CoV-2 Spike- and EBV-specific CD3^+^, CD4^+^ and CD8^+^ T cells in Allo-HSCT recipients (*n* = 19 [P] and 12 healthy donors [HD]). Data are shown as means of the percentage of T-cell responders.

**Figure 3 vaccines-10-00448-f003:**
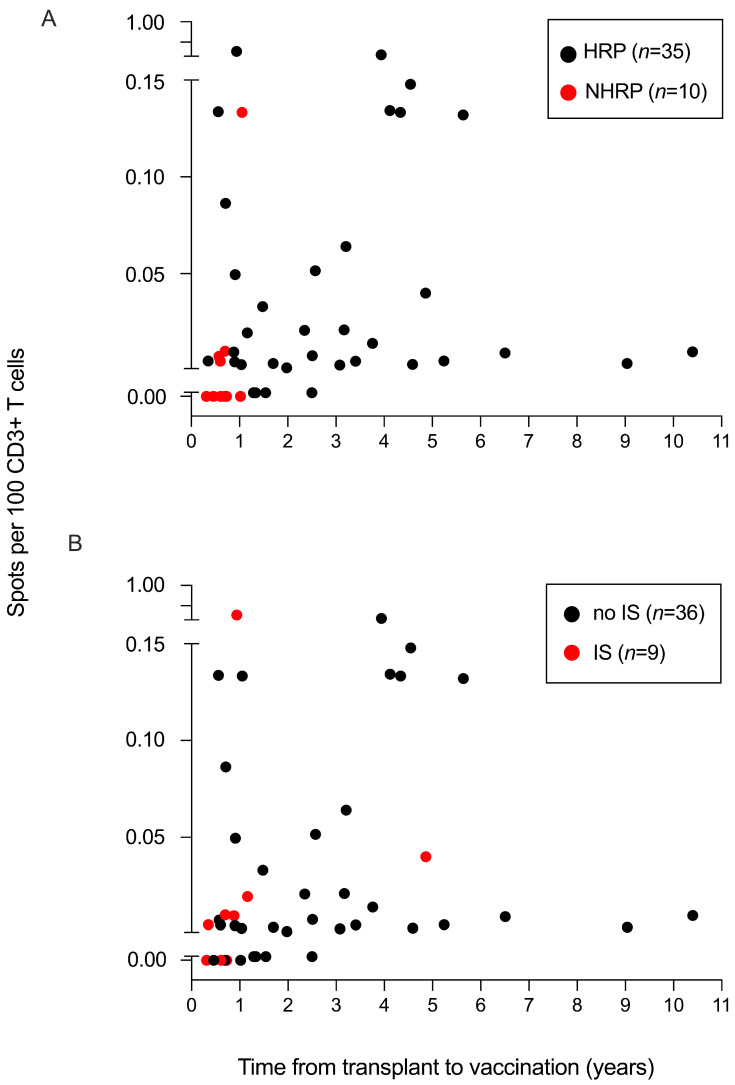
Anti-Spike T-cell response according to the time interval between transplantation and vaccination. (**A**) Anti-Spike T-cell responses according to the time interval between transplantation and vaccination for non-humoral responder patients (NHRP, *n* = 10) and humoral responder patients (HRP, *n* = 35). (**B**) Anti-Spike T-cell responses according to the time interval between transplantation and vaccination for patients under immunosuppressive therapy (IS, *n* = 9) or not (*n* = 36, 80%) at the time of vaccination. Patients were under treatment including 5 for active chronic GVHD (cyclosporine *n* = 2, cyclosporine + corticosteroids *n* = 3), while one patient was under corticosteroids for a chronic rheumatic disease and another one received 5′ azacytidine for relapse prevention. Two early patients were on their way to stop cyclosporine. None of the patients had acute graft-versus-host disease (GVHD) at the time of analysis.

**Table 1 vaccines-10-00448-t001:** Vaccinated individuals’ characteristics.

	Allogeneic Hematopoietic Stem-Cell Recipients (*N* = 45)				Healthy Controls (*N* = 16)
Antibody response after two doses of BNT162b2 vaccination	Yes (HR) ^a^ 35 (78%)		No (NHR) ^b^ 10 (22%)		Yes 16 (100%)
T-cell response after two doses of BNT162b2 vaccination	Yes 31 (89%)	No 4 (11%)	Yes 4 (40%)	No 6 (60%)	Yes 16 (100%)
Median time from transplant to vaccination (days)	1026	523	236	237	NA
Range	(126–3796)	(471–914)	(208–384)	(112–372)	
Median time from first to second vaccination (days)	21	28	23	21	24
Range	(19–35)	(21–29)	(16–29)	(21–29)	(18–32)
Median time from second vaccination to T cells response analyses (days)	32	36	62	45	58
Range	(22–67)	(25–69)	(56–70)	(26–56)	(32–70)
Median time from first vaccination to T cells response analyses (days)	56	64	85	66	81
Range	(43–95)	(52–90)	(85–86)	(47–85)	(62–91)
Underlying disease	18 AML 13 MDS	4 MDS	3 AML 1MDS	5 AML 1MDS	NA
Median age: years	62	58	66	57	52
(range)	(30–75)	(49–72)	(41–70)	(44–66)	(37–63)
Gender					
Male	18	2	2	4	3
Female	13	2	1	2	13
Donor type					
Geno-identical	6	1	1	0	NA
MUD	15	2	0	2	
Haploidentical	9	1	3	4	
9/10 mis-MUD	1	0	0	0	
Conditioning					
Myeloablative	1	0	0	0	NA
Reduced-intensity	29	4	4	6	
Sequential	1	0	0	0	
GVHD prophylaxis					
CsA + MMF + ATG	15	1	0	1	NA
CsA + MMF + PTCY	7	3	4	5	
PTCY only	10	0	0	0	
Previous GVHD					
Yes	19 (61%)	2 (50%)	2 (50%)	3 (50%)	NA
No	12 (39%)	2 (50%)	2 (50%)	3 (50%)	
Ongoing treatment *					
No	26 (84%)	4 (100%)	3 (75%)	3 (50%)	NA
Yes	5 (16%)	0	1 (25%)	3 (50%)	

^a^: Humoral Responders. ^b^: Non Humoral Responders. AML: acute myeloid leukemia; MDS: myelodysplastic syndrome; MUD: matched unrelated donor; GVHD: graft-versus-host disease; CsA: cyclosporine; MMF: mycophenolate mofetyl; PTCY: post-transplant Cyclophosphamide; NA: not applicable. *: immunosuppressive drugs or chemotherapy.

## Data Availability

The data are available by contacting corresponding authors.

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
