# Peer review of "SARS-CoV-2 T-Cell Responses in Allogeneic Hematopoietic Stem Cell Recipients following Two Doses of BNT162b2 mRNA Vaccine"

_vaccines, 2022, doi:10.3390/vaccines10030448_

Round 1

Reviewer 1 Report

Authors reported the simple statistic analysis data regarding to more than 100 Aollo-HSCT patients whose T cell responses after two shots of BNT mRNA vaccine. Overall, it is a good report, although it is not a novelty, it is really one of the pioneer studies. The weak antibody immuno-response tis known in Allo-HSCT recipients, and the role of T cell role of the immune system is brought into sharper focus. T cells might be overlooked in the fight against COVID-19. This report may provide an interested and useful information. As if authors can provide more discussions among the 4 similar reports, including authors’ report, that should improve the importance of this manuscript. For a brief report point of view, I only have two more minor comments:

  1. Authors should carefully check whether there are more typing errors in the text including figure legends, such as: 1). Page 5 line 155 and line 157, the “(a)” and “(b)” should be corrected as “(A)” and “(B)”, respectively, to appropriately correlate figure label with its relative legend. 2). Page 7 line 188, …. (data not shown). “(D)” Bar graphs showing…. That should be a typing error and to be corrected as “(C)”.
  2. For figure 1 & 2, the small font size and its low resolution that make not clearly enough labelling for the graphs.

Author Response

Reviewer 1: 

Authors reported the simple statistic analysis data regarding to more than 100 Aollo-HSCT patients whose T cell responses after two shots of BNT mRNA vaccine. Overall, it is a good report, although it is not a novelty, it is really one of the pioneer studies. The weak antibody immuno-response tis known in Allo-HSCT recipients, and the role of T cell role of the immune system is brought into sharper focus. T cells might be overlooked in the fight against COVID-19. This report may provide an interested and useful information. As if authors can provide more discussions among the 4 similar reports, including authors’ report that should improve the importance of this manuscript.

We have now discussed the specific T-cell responses in terms of CD4/CD8 + T cells responses comparing our results with those published. We have also provided a paragraph regarding predictive factors of the humoral and cellular responses.

For a brief report point of view, I only have two more minor comments:

  1. Authors should carefully check whether there are more typing errors in the text including figure legends, such as: 1). Page 5 line 155 and line 157, the “(a)” and “(b)” should be corrected as “(A)” and “(B)”, respectively, to appropriately correlate figure label with its relative legend. 2). Page 7 line 188, …. (data not shown). “(D)” Bar graphs showing…. That should be a typing error and to be corrected as “(C)”.

Changes have been made accordingly.

  1. For figure 1 & 2, the small font size and its low resolution that make not clearly enough labelling for the graphs.​

Figures are now provided in the .jpg format showing adequate resolution.

Reviewer 2 Report

This is a well performed and well written study on the occurrence and function of SARS-Cov-2 spike protein-specific CD4 T cells after vaccination in patients with hematopoietic stem cell transplantation. The study is of high interest in general and for patients with HSCT.

Concerns

-Graphic quality of figures is poor and has to be improved.

Author Response

Reviewer 2: This is a well performed and well written study on the occurrence and function of SARS-Cov-2 spike protein-specific CD4 T cells after vaccination in patients with hematopoietic stem cell transplantation. The study is of high interest in general and for patients with HSCT.

Concerns

-Graphic quality of figures is poor and has to be improved.

Figures are now provided in the .jpg format showing adequate resolution.
